# An Inhibitory Role for Human CD96 Endodomain in T Cell Anti-Tumor Responses

**DOI:** 10.3390/cells12020309

**Published:** 2023-01-13

**Authors:** Chelsia Qiuxia Wang, Fong Chan Choy, Arleen Sanny, Takashi Murakami, Andy Hee-Meng Tan, Kong-Peng Lam

**Affiliations:** 1Bioprocessing Technology Institute, Agency for Science, Technology and Research, 20 Biopolis Way, #06-01 Centros, Singapore 138668, Singapore; 2Department of Microbiology, Saitama Medical University, Moroyama, Saitama 350-0495, Japan; 3Singapore Immunology Network, Agency for Science, Technology and Research, Singapore 138648, Singapore; 4Department of Physiology, Yong Loo Lin School of Medicine, National University of Singapore, Singapore 117545, Singapore; 5Department of Microbiology and Immunology, Yong Loo Lin School of Medicine, National University of Singapore, Singapore 117545, Singapore; 6Department of Pediatrics, Yong Loo Lin School of Medicine, National University of Singapore, Singapore 117545, Singapore; 7School of Biological Sciences, Nanyang Technological University, Singapore 637551, Singapore

**Keywords:** CD96, immune checkpoint, immunotherapy, immunologic receptors, T cell cytotoxicity

## Abstract

Immune checkpoint blockade (ICB) therapy involves the inhibition of immune checkpoint regulators which reverses their limitation of T cell anti-tumor responses and results in long-lasting tumor regression. However, poor clinical response or tumor relapse was observed in some patients receiving such therapy administered via antibodies blocking the cytotoxic T lymphocyte-associated protein 4 (CTLA-4) or the programmed cell death 1 (PD-1) pathway alone or in combination, suggesting the involvement of additional immune checkpoints. CD96, a possible immune checkpoint, was previously shown to suppress natural killer (NK) cell anti-tumor activity but its role in human T cells remains controversial. Here, we demonstrate that CRISPR/Cas9-based deletion of *CD96* in human T cells enhanced their killing of leukemia cells in vitro. T cells engineered with a chimeric antigen receptor (CAR) comprising human epidermal growth factor receptor 2 (EGFR2/HER2)-binding extracellular region and intracellular regions of CD96 and CD3ζ (4D5-96z CAR-T cells) were less effective in suppressing the growth of HER2-expressing tumor cells in vitro and in vivo compared with counterparts bearing CAR that lacked CD96 endodomain (4D5-z CAR-T cells). Together, our findings implicate a role for CD96 endodomain in attenuating T cell cytotoxicity and support combination tumor immunotherapy targeting multiple rather than single immune checkpoints.

## 1. Introduction

Targeting immune checkpoint regulators in patients with various malignancies has led to unprecedented success in oncologic therapy [1,2,3,4]. Immune checkpoints, collectively referring to the myriad of co-inhibitory signaling molecules expressed by immune cells, serve to dampen the activation of T and natural killer (NK) cells and prevent their indiscriminate attack on host cells to maintain self-tolerance. Tumor cells are known to hijack this fail-safe mechanism by stimulating these checkpoints to avert immune attack. Past studies have unravelled possible mechanisms underlying immune checkpoint blockade (ICB) therapy to be reversal of termination of T cell priming and/or rejuvenation and expansion of exhausted tumor-specific T cells, thereby restoring T cell anti-tumor activity [3,4].

In this regard, anti-programmed cell death 1 (PD-1)/PD-L1 and anti-cytotoxic T lymphocyte-associated protein 4 (CTLA-4) therapies have been demonstrated to elicit dramatic tumor regression in some patients [4]. Unfortunately, a significant proportion of patients do not respond adequately or ultimately show resistance to these treatments, motivating the continued search for additional immune checkpoint receptors that can be therapeutically targeted. One recent candidate whose function has been under intense scrutiny is CD96, a receptor which is almost exclusively expressed in NK cells and in certain subpopulations of T and B cells [5,6].

CD96 is a member of the immunoglobulin (Ig) superfamily that interacts with nectin and nectin-like proteins [7]. Other members include DNAX Accessory Molecule-1 (DNAM-1/CD226) and *T*-cell immunoreceptor with *Ig* and immunoreceptor tyrosine-based inhibition motif (ITIM) domains (TIGIT/WUCAM/VSTM3). Although CD96 was discovered and cloned more than two decades ago as *T* cell activation antigen, increased late expression (TACTILE) [5], its role in immune checkpoint regulation was not appreciated until recently, following the identification of CD155 (PVR/NECL-5) and CD111 (Nectin-1) as its ligands [4,5,6]. CD96 competes with CD226 and TIGIT for binding to CD155 with an affinity exceeding that of CD226 but lower than that of TIGIT [8].

Previous studies have unequivocally demonstrated that CD96 plays an inhibitory role in the anti-tumor responses of murine NK cells. Using *CD96*-deficient mice, some of these studies demonstrated that CD96 limited the production of interferon-γ (IFN-γ) by NK cells, but not their cytotoxic function, the former being important for tumor control in two independent models of carcinogenesis and metastasis [9]. In contrast to murine CD96 which harbors an ITIM motif in its cytoplasmic domain (endodomain), human CD96 harbors both an ITIM motif and an activation-associated YXXM motif (SH2 domain binding site), suggesting it can be co-inhibitory or co-stimulatory depending on operational context [10]. Although an activating role for CD96 in human NK cells has been reported [6,11,12], a recent study observed that intratumoral NK cells in hepatocellular carcinoma (HCC) patients with high CD96 expression were functionally exhausted and compromised in the production of anti-tumor cytokines IFN-γ and tumor necrosis factor-α (TNF-α). High CD96 expression was also found to be strongly correlated with poor patient disease-free survival [13]. Notwithstanding the lack of a mechanistic investigation of how CD96 inhibited cytokine production and tumor killing by NK cells, these findings, together with those described for mouse NK cells, suggest that CD96 negatively regulates the anti-tumor responses of mouse and human NK cells.

Whether CD96 plays an inhibitory or stimulatory role in T cells remains largely conflicting. In support of the former, CD96 was found to be co-expressed with TIGIT and/or PD-1 in mouse and human tumor-infiltrating lymphocytes (TILs). Moreover, selectively blocking CD96 alone or in combination with PD-1/PD-L1 enhanced control of tumors in several experimental mouse models in a CD8^+^ T cell-dependent manner [14]. On the contrary, in support of a costimulatory role for CD96, the crosslinking of CD96 on mouse and human CD8^+^ T cells was found to induce T cell activation, proliferation and cytokine production in part via the MEK-ERK pathway. CD96 deficiency reduced the frequencies of NUR77-, T-bet-, TNFα- and IFNγ-expressing tumor-specific CD8^+^ T cells in a mouse model of colorectal carcinoma, suggesting CD96 co-stimulated rather than inhibited effector T cell anti-tumor responses [15]. The role of CD96 was further complicated by another study which showed that antibody (Ab)-mediated blockade of CD96 did not increase IFN-γ secretion by CD8^+^ T cells, suggesting CD96 does not suppress T cell function [16].

In this study, we attempted to clarify the role of CD96 in T cells by knocking out *CD96*, as opposed to Ab blockade, in human peripheral blood-derived T cells and engineering human T cells expressing a CAR incorporating CD96 endodomain. We showed that genetic ablation of *CD96* in human T cells enabled these cells to kill chronic myeloid leukemia (CML) cells and a subset of acute myeloid leukemia (AML) cells more efficiently in vitro. Furthermore, T cells engineered with a chimeric antigen receptor (CAR) containing a single chain variable fragment (scFv) based on the monoclonal Ab (mAb) clone 4D5 that recognizes human epidermal growth factor receptor 2 (EGFR2/HER2) antigen and the endodomains of CD96 and CD3ζ (4D5-96z CAR-T cells) killed HER2-expressing tumor cells less efficiently in vitro and in vivo than T cells engineered with a CAR containing the same scFv and the endodomain of CD3ζ alone (4D5-z CAR-T cells). Collectively, these findings suggest that the endodomain of CD96 has inhibitory function in human T cells.

## 2. Materials and Methods

### 2.1. Primary Cells and Cell Lines

Luciferase-expressing SK-BR-3 (#JCRB1627.1, SK-BR-3-Luc) and SK-OV-3 (#JCRB1594, SK-OV-3/CMV-Luc) cells were provided by the *Japanese Collection of Research Bioresources* (JCRB) Cell Bank (Osaka, Japan). K562 cells were a kind gift of Dr Shu Wang (National University of Singapore). MOLM14, U937 and MV4;11 cells were a kind gift of Dr Alice Cheung (Singapore General Hospital). K562, MOLM14, U937 and MV4;11 cells were engineered to express luciferase using lentiviral transduction with pLenti CMV Puro LUC [17], a gift from Eric Campeau & Paul Kaufman (Addgene, Watertown, MA, USA, #17477). Transduced cells were placed under puromycin (InvivoGen, San Diego, CA, USA, #ant-pr) selection for at least 1 week. All cell lines were cultured in medium consisting of RPMI 1640 (Nacalai Tesque, Kyoto, Japan, #30264-56) supplemented with 10% heat-inactivated fetal bovine serum (FBS, HyClone Logan, Utah, USA, #SV30160.03) and 1% penicillin/streptomycin (Gibco, Thermo Fisher Scientific, Waltham, MA, USA, #15140-122) at 37 °C and in 5% CO_2_. Frozen human peripheral blood mononuclear cells (PBMCs) from various de-identified donors were purchased from STEMCELL Technologies, Vancouver, Canada, USA. PBMCs were thawed in a 37 °C water bath and washed twice in pre-warmed complete RPMI 1640 supplemented with 10% FBS and 1% penicillin/streptomycin. Cell viability and density were determined using 0.2% w/v Trypan blue solution in phosphate buffered saline (PBS) (Sigma, St. Louis, MO, USA, #T6146).

### 2.2. CRISPR/Cas9-Mediated Gene Deletion in T Cells

PBMCs were activated in the presence of 50 ng/mL soluble anti-CD3 (OKT3; eBioScience, Thermo Fisher Scientific, Waltham, MA, USA, #16-0037-85) and 100 ng/mL anti-CD28 (CD28.2; eBioScience, Thermo Fisher Scientific, Waltham, MA, USA, #16-0289-85) Abs, 20 IU/mL recombinant human interleukin (IL)-2 (Peprotech, London, UK, #200-02-1000/#200-02-500), 5 ng/mL recombinant human IL-7 (STEMCELL Technologies, Vancouver, Canada, #78053.2) and 5 ng/mL recombinant human IL-15 (STEMCELL Technologies, Vancouver, Canada, #78031.2) for 3 days. To generate *CD96* knockout (KO) T cells following activation, guide RNAs (gRNAs) targeting the *CD96* locus and Alt-R^®^ *Streptococcus pyogenes* (S.p.) HiFi Cas9 Nuclease V3 protein (Integrated DNA Technologies, IDT, Coralville, Iowa, USA, #1081060) were pre-incubated and introduced as ribonucleoprotein (RNP) complex into T cells via electroporation using P3 Primary cell 4D-Nucleofactor X Kit (Lonza, Basel, Switzerland, V4XP-3032/V4XP-3024) and program EH-115 in 4D-Nucleofector X Unit (Lonza, Basel, Switzerland). Electroporated cells were then expanded in the presence of 100 IU/mL IL-2 for further 3 days. AA and AB were gRNAs designed by IDT to target *CD96* exon 2 while AC gRNA was designed to target exon 3. Sequence 3, 5 and 6 gRNAs were designed in-house to target the transcription start site (TSS) of *CD96* exon 1 (Appendix A; sequence 3: 5′-TCGGGTTTTTTAGCACGAAG-3′; sequence 5: 5′-CGGGTTTTTTAGCACGAAGT-3′; sequence 6: 5′-CGAACTCTACCACACACGCC-3′). To generate *CD96* wild-type (WT) T cells, T cells were mock electroporated with Cas9 and tracrRNA alone.

### 2.3. Retroviral CAR Construction

CARs contain HER2-binding scFv moiety 4D5 sequence derived from the humanized mAb Herceptin (Trastuzumab) [18]. CD8α hinge and transmembrane region and CD3ζ intracellular signaling domain (IC) alone (4D5-z) or in tandem with CD96 IC (4D5-96z) were synthesized (Bio Basic Asia Pacific Pte Ltd, Singapore) and sub-cloned in frame into MSGV-Hu-Acceptor retroviral vector [19], a gift from David Ott (Addgene #64269), after removal of human TCR genes. Details of sequences are available upon request.

### 2.4. Retroviral Vector Production and Generation of CAR-T Cells

Phoenix-GP cells stably expressing MoMLV gag-pol (ATCC, Manassas, Virginia, USA, #CRL-3215) were co-transfected with MSGV-based CAR constructs and pCMV-VSV-G [20], a gift from Bob Weinberg (Addgene, Watertown, MA, USA, #8454), using FuGENE 6 (Promega, Madison, WI, USA, #TM350). Retroviral supernatant was collected 48 h after transfection, filtered with a 0.45-μm filter, and used to transduce PG13 cells (ATCC, Manassas, Virginia, USA, #CRL-10686) by spinoculation at 600× *g* for 2 h to generate cells which produce GaLV-pseudotyped CAR retrovirus (CAR PG13 cells) and cryopreserved. One week before activation of PBMCs, CAR PG13 cells were thawed and cultured to generate viral supernatant which was filtered, added and centrifuged at 1500× *g* for 2 h to bind RetroNectin (rFN-CH-296; Takara Bio Inc., Shiga, Japan #T100B) pre-coated at 5.26 μg/cm^2^ in non-tissue culture-treated 6-well plates. PBMCs were seeded as described above for CRISPR/Cas9 experiments without IL-7 and IL-15 for 2 days and activated T cells were transduced with CAR or not by applying them to CAR or mock virus-bound RetroNectin, respectively, at 600× *g* for 30 min. Transduced T cells were expanded in the presence of 100 IU/mL IL-2 for further 3–6 days before use. CAR expression in PG13 and T cells was routinely assessed by flow cytometry to be >90% and 40%–70% respectively.

### 2.5. Flow Cytometry

Before staining with relevant fluorochrome-conjugated Abs, cells were treated with Human TruStain F_c_X (F_c_ receptor blocking solution; BioLegend, San Diego, CA, USA, #422302). Abs against CD4 (APC; OKT4; #317416), CD8 (PerCP-Cy5.5; HIT8a; #300924), TIGIT (PE-Cy7; A15153G; #372713), CD226 (FITC; 11A8; #338303), CD155 (Alexa Fluor 647; SKII.4; #337621), CD111 (PE; R1.302; #340404), HER2 (PE; 24D2; #324405) and mouse IgG_1_ κ isotype control (PE; MOPC-21; #400113) were from BioLegend, San Diego, CA, USA. Ab against CD96 (PE; NK92.39) was from eBioscience, Thermo Fisher Scientific, Waltham, MA, USA (#12-0969-42) or BioLegend, San Diego, CA, USA (#338405). Cell surface expression of CAR was detected by incubation with recombinant biotinylated protein L (ACROBiosystems, Newark, DE, USA, #RPL-P814R) followed by incubation with PE-conjugated streptavidin (eBioscience, Thermo Fisher Scientific, Waltham, MA, USA, #12-4317-87). To exclude dead cells, 4′,6-diamidino-2-phenylindole dihydrochloride (DAPI; Biolegend, San Diego, CA, USA, #422801) was used. Samples (at least 20,000 events) were acquired on a LSRII cytometer (BD Biosciences) or MACSQuant X (Miltenyi Biotec) and analyzed with FlowJo software (TreeStar, BD Life Sciences, Franklin Lakes, NJ, USA).

### 2.6. In Vitro Cytotoxicity Assay

T cells were co-cultured with a fixed number (1.125–2.25 × 10^4^) of luciferase-expressing K562, MOLM14, U937 or MV4;11 cells at effector (E; T cell) to target (T; tumor cell) ratios ranging from 1:1 to 20:1 in 96-well plate for 20 h. The luciferase-catalyzed bioluminescence (BLI), in absolute luciferase units, of surviving tumor cells was assessed via the Bright-Glo Luciferase Assay System (Promega, Madison, WI, USA, #E2620) following the manufacturer’s instructions. Hence, 75 μL of culture medium was mixed with 75 μL of the prepared luciferase reagent in each well and the plates were shaken for 5 min to allow complete lysis of cells. Luminescence of the lysed mixture was measured using the Synergy HTX Multi-Mode Microplate Reader (BioTek Winooski, VT, USA). Percentage (%) cytotoxicity was calculated as: Δluc [luc (no T cells) – luc (T cells)] / luc (no T cells) × 100%, where luc represents absolute luciferase units. A similar procedure was carried out for experiments using luciferase-expressing SK-BR-3 and SK-OV-3, with the following modifications: T cells were co-cultured with 3 × 10^4^ tumor cells and wells were emptied before the addition of 2 × PBS-diluted luciferase reagent.

### 2.7. Xenograft Mouse Models

Briefly, 6–8-week-old female NSG (NOD.Cg-*Prkdc^scid^Il2rg^tm1Wjl^*/SzJ) mice (The Jackson Laboratory stock #005557) were purchased from InVivos, Singapore and injected intraperitoneally with 7 × 10^6^ luciferase-expressing SK-BR-3 cells or 1.5 × 10^6^ luciferase-expressing SK-OV-3 cells per mouse. After 3 weeks to allow for successful SK-BR-3 engraftment, mice were randomly distributed into three groups and infused with 7 × 10^6^ non-CAR, 4D5-z or 4D5-96z CAR-T cells per mouse at E:T ratio of 1:1. SK-OV-3-injected mice were rested for 8 days to allow for tumor engraftment and subsequently infused with 1.5 × 10^6^ non-CAR, 4D5-z or 4D5-96z CAR-T cells per mouse at E:T ratio of 1:1. Mice which did not show tumor engraftment prior to T cell infusion were not included in the study. Tumor burden was monitored weekly by BLI using an IVIS Spectrum In Vivo Imaging System (PerkinElmer, Waltham, MA, USA). Average radiance, defined as the sum of the radiance from each pixel inside the region of interest (ROI)/number of pixels or super pixels (photons/s/cm^2^/steradian (sr)), was calculated for each mouse using the Living Image^®^ Software (PerkinElmer, Waltham, MA, USA). Experiments with mice were approved by the Institutional Animal Care and Use Committee (IACUC) at BRC, A*STAR. We used the ARRIVE checklist when writing our report [21].

### 2.8. Statistical Analyses

Differences in numerical values between samples used in in vitro cytotoxicity assays were compared using multiple unpaired Student’s *t*-test (for parametric data sets with 2 groups) or two-way ANOVA with Tukey *post hoc* analysis (for parametric data sets with 3 groups). Differences in average radiance output of in vivo BLI imaging and ratios of tumor-bearing mice were compared using Kruskal-Wallis test (for non-parametric data sets with 3 groups) and Fisher exact test, respectively. In all tests, a value of *p* < 0.05 for a given comparison determined by Prism (version 8; GraphPad Software, San Diego, CA, USA) was regarded as statistically significant.

## 3. Results

### 3.1. CRISPR/Cas9-Mediated Deletion of the CD96 Gene in Human T Cells

The role of CD96 in T cells is not well defined. To investigate how CD96 regulates the anti-tumor activity of human T cells, we adopted a CRISPR/Cas9 approach to delete the *CD96* gene in peripheral blood-derived T cells. Cells were activated with anti-CD3 and anti-CD28 Abs for three days, following which CD96 expression was assessed by flow cytometry. We confirmed that CD96 is expressed in both activated CD4^+^ and CD8^+^ T cells, as well as in their naïve counterparts before activation (Figure 1A). Activation of T cells was routinely assessed via microscopic examination for T cell clustering and/or flow cytometric analysis of CD69 expression in T cells (Figure 1B). We next transfected T cells with no gRNA (mock-electroporated) to generate WT T cells or with one of six gRNAs designed to target exons 1, 2 or 3 of the *CD96* locus which are shared by currently annotated *CD96* transcripts to generate *CD96* KO T cells (Appendix A). Transfected cells were then cultured for three days and the abrogation of CD96 protein expression was verified by flow cytometry (Appendix A). AA, AB, and AC gRNAs mediated an almost complete deletion of CD96. Sequence 5 gRNA mediated-editing reduced CD96 expression by half whereas sequence 3 and 6 gRNAs affected CD96 expression negligibly. *CD96* deletion occurred with similar efficiency in CD4^+^ and CD8^+^ T cells (Appendix A) and electroporation *per se* did not further alter CD96 expression in T cells (Appendix A, mock-electroporated *CD96* WT compared with non-electroporated). CD96 expression was progressively upregulated in T cells over six days in culture after initial activation, consistent with previous reports [5,22].

### 3.2. CD96 Deletion in T Cells Enhances T Cell Cytotoxicity against K562 CML Cells

Interaction of CD96 with CD155 has been shown to mediate the cytotoxic function of NK cells against CD155-expressing target cells [4]. In addition, Abs targeting CD96 that failed to block CD96–CD155 interactions nonetheless enabled NK cells to suppress experimental and spontaneous metastases [23] suggesting the involvement of other CD96 ligands, e.g., CD111, in mediating NK anti-tumor activity. We assessed and found that the majority of K562 CML (78.5%), MOLM14 AML (99.5%), U937 AML (99.9%), MV4;11 AML (94.8%), SK-BR-3 metastatic breast adenocarcinoma (94.1%) and SK-OV-3 ovarian adenocarcinoma cells (99.2%) were CD155^+^, while levels of CD111 expression varied widely with tumor type: 0.14% (hardly detectable) of K562, 12.1% of MOLM14, 89.7% of U937, 13.8% of MV4;11, 89.5% of SK-BR-3, and 18.2% of SK-OV-3 cells were CD111^+^ (Figure 2A). To examine how *CD96* deficiency in T cells and differential CD111 expression in these tumor cells modulated the cytotoxic capacity of T cells against tumor cells, we co-incubated *CD96* WT or KO T cells with luciferase-expressing tumor cells at different E:T ratios for 20 h. We observed that *CD96* deficiency in T cells edited using gRNAs was shown to achieve high levels of gene deletion (Appendix A) with increased T cell killing of K562 cells at most E:T ratios as assessed by BLI assay (Figure 2B). *CD96* deletion in T cells also enhanced T cell killing of MOLM14 cells at lower E:T ratios and U937 cells at most E:T ratios examined, although increase in killing of the latter cells did not reach statistical significance (Figure 2C). In contrast, the loss of *CD96* in T cells did not accentuate their killing of MV4;11, SK-BR-3, and SK-OV-3 cells regardless of E:T ratio and donor source of T cells (Figure 2D,E). The lack of apparent correlation between expression of known CD96 ligands on target cells and CD96-mediated functional inhibition raises the possibility that ligands apart from CD155 and CD111 mediate the functional effects of CD96 in T cells. In addition, the modest increase of T cell cytotoxicity against K562 resulting from *CD96* deficiency was not due to compensatory increase in expression of the co-inhibitory receptor TIGIT, which competes with CD96 for binding to CD155, to mitigate exacerbation of T cell cytotoxicity mediated by the absence of CD96. This was because *CD96* deletion did not affect TIGIT expression in T cells (Appendix A, lower panel). Moreover, CD226 remained highly expressed following abrogation of *CD96* in T cells (Appendix A, upper panel). Taken together, our results suggest that CD96 suppresses the cytotoxicity of T cells against a subset of CD155-expressing tumor cell types (Appendix A).

### 3.3. T Cells Expressing Anti-HER2 CAR Incorporating CD96 Endodomain Modestly Attenuated Cytotoxic Function In Vitro and In Vivo

We next aimed to decipher the intracellular signaling biology of CD96 and, at the same time, avoid the complexity of ligands triggering the CD96 native receptor confounding interpretation. To this end, we adopted a complementary approach whereby T cells were engineered to express a CAR comprising an extracellular scFv based on the mAb trastuzumab (Herceptin) clone 4D5 [18] and either the endodomain of CD3ζ alone (4D5-z CAR-T cells) or in combination with the endodomain of CD96 (4D5-96z CAR-T cells; Figure 3A). Trastuzumab recognizes EGFR2/HER2 overexpressed in many tumor types, including SK-BR-3 and SK-OV-3 (Figure 3B). We co-cultured CAR-T cells with luciferase-expressing SK-BR-3 and SK-OV-3 cells at various E:T ratios and assessed BLI after 20 h. Consistent with suppression of cytotoxic function by CD96 in T cells, 4D5-96z CAR-T cells were found to be less cytotoxic compared with 4D5-z counterparts against tumor cells highly expressing HER2 (Figure 3C). Importantly, this was not due to 4D5-z and 4D5-96z CAR-T cells bearing different frequencies of CAR^+^ populations (Figure 3D). These observations were reproducible with CAR-T cells generated from at least two independent donors (Appendix A).

As further validation of our preceding findings, we assessed the capacity of 4D5-96z versus 4D5-z CAR-T cells to control the growth of HER2-expressing SK-BR-3 tumor cells in vivo. We injected SK-BR-3 cells intraperitoneally into immunodeficient NSG mice which were left for three weeks to establish tumor growth (Figure 4A). The mice were subjected to BLI imaging prior to CAR-T infusion to confirm the comparable extent of tumor engraftment in all mice, and thereafter once every week to monitor tumor progression. They were randomly assigned to receive either control T cells expressing no CAR (non-CAR-T), 4D5-z or 4D5-96z CAR-T cells (Figure 4B,C, day 0). We confirmed similar frequencies of CAR^+^ cells within 4D5-z and 4D5-96z CAR-T cells (data similar to Figure 3D not shown) before they were infused into tumor-bearing mice. As expected, SK-BR-3 tumors were extinguished by day 14 in all mice infused with 4D5-z CAR-T cells (n = 6) and they remained in remission at day 28. In contrast, tumors were eliminated in some but failed to be resolved in other mice that received 4D5-96z CAR-T cells at the corresponding time points examined (Figure 4B,C, day 14 and day 28; Figure 4D, day 14). We repeated the above experiment by replacing SK-BR-3 with SK-OV-3 tumor cells which yielded similar results, although statistical significance was not reached (Appendix A). Overall, our data demonstrate that CD96 signaling inhibits CAR-mediated CD3ζ activation in human T cells.

## 4. Discussion

Previous studies have unequivocally demonstrated that CD96 plays an inhibitory role in the anti-tumor responses of murine NK cells [9,24,25,26]. The function of CD96 in T cells is far less clear. In this brief report, we showed that CD96 inhibits human T cell anti-tumor cytotoxicity in vitro and in vivo. Activated T cells from human peripheral blood that were edited to lack the *CD96* gene exhibited enhanced killing of K562 CML and MOLM14 AML tumor cells expressing the CD96 ligand CD155 compared with mock-edited control cells. Consistent with this finding, the cytotoxicity of T cells engineered to express an anti-HER2 CAR containing CD96 and CD3ζ endodomains against HER2-expressing tumor cells was significantly impaired as compared with counterparts expressing an anti-HER2 CAR containing CD3ζ endodomain alone. This complementary strategy using HER2-targeting CAR was aimed at deciphering the intracellular signaling mediated by CD96 independent of the complexity of ligands triggering the CD96 native receptor which may confound interpretation. The modest but significant inhibition of T cell cytotoxicity supports earlier evidence that blocking CD96 in combination with other checkpoints promoted robust tumor control by murine CD8^+^ T cells, while blocking CD96 alone resulted in minimal tumor control by T cells [14].

Contradictory to the suppressive function of CD96 elucidated by our and prior published work [14], a recent study ascribed a co-stimulatory function for CD96 in both mouse and human CD8^+^ T cells [15]. To draw this conclusion, the authors compared crosslinking CD3 and CD96 via beads coupled with anti-CD3 and anti-CD96 Abs versus crosslinking CD3 and CD28 via beads coupled with anti-CD3 and anti-CD28 Abs to activate CD8^+^ T cells in vitro. They found that cells activated by either bead type proliferated and increased the expression and/or phosphorylation of various signaling molecules including transcription factors to similar extent. Treatment with soluble anti-CD96 Ab alone did not induce activation and proliferation of CD8^+^ T cells. Curiously, stimulation with beads coated with anti-CD3 and anti-CD226 Abs was also found to induce ERK phosphorylation which persisted over time in *CD96* KO but not WT murine CD8^+^ T cells, suggesting CD96 dampened T cell activation via CD3 and CD226 engagement.

In our study, we sought to address how CD96 could affect the anti-tumor activity of T cells by generating *CD96* KO T cells. We found that while *CD96* deficiency in T cells moderately crippled their ability to kill K562 and MOLM14 cells, abrogation of *CD96* did not affect T cell cytotoxicity against other cell lines tested. Another study documented that Ab-mediated antagonism of CD96 in CD8^+^ T cells did not augment their cytokine production and cytotoxic activity, a discrepancy with part of our data which may be explained by our analysis of CD96 function in total instead of CD8^+^ T cells. Further investigation is needed to clarify the specific contexts in which human CD96 serves an inhibitory or stimulatory role in T cell cytotoxicity.

The limitation of T cell killing by CD96 may apply to selected tumor types. Surprisingly, we found that whereas killing of K562, MOLM14, and U937 cells by *CD96* KO T cells was increased, killing of MV4;11, SK-BR-3, and SK-OV-3 cells by *CD96* KO T cells was unchanged compared with WT counterparts, despite MV4;11, SK-BR-3, and SK-OV-3 cells co-expressing CD155 and CD111. This suggests that additional CD96 ligands may be present on tumor cells that mediate CD96 suppression, the identification of which is essential to fully elucidate CD96 function.

## 5. Conclusions

In summary, our data suggest that the endodomain of CD96 transduces inhibitory signaling that dampens CD3ζ activation mediated by CAR in human T cells (Appendix A) and provide the basis for future interrogation of the relative contributions of signaling motifs residing within the endodomain to the capacity of T cells to eradicate tumors. These insights can be exploited as part of combination immunotherapy to improve the efficacy of T cell anti-tumor responses. Since anti-PD-1/PD-L1 or anti-CTLA4 monotherapy is known to benefit some but not all patients, we expect that combination therapy including CD96 blockade will enable a wider cohort of patients to benefit from ICB therapy, especially when additionally bolstered with non-ICB CAR-T therapy.

## Figures and Tables

**Figure 1 cells-12-00309-f001:**
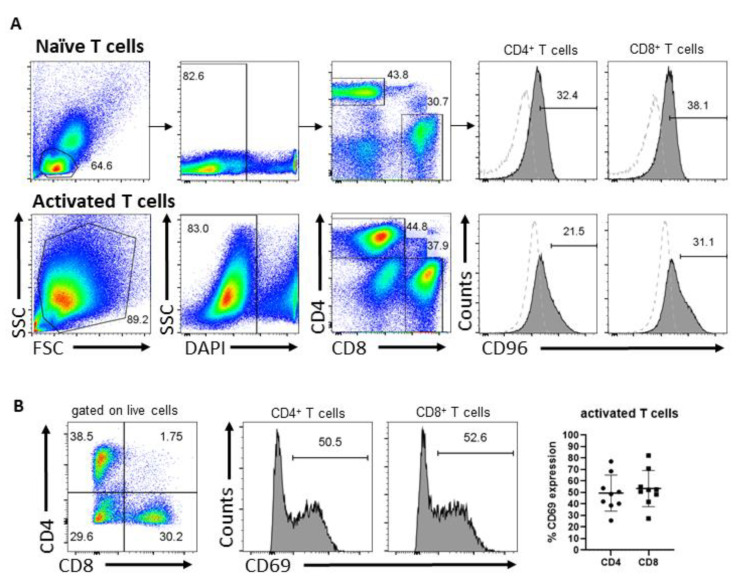
CD96 and CD69 expression in T cells. (**A**) Expression of CD96 in naïve CD4^+^ and CD8^+^ T cells at day 0 (top) and T cells activated by anti-CD3 and anti-CD28 Abs for 3 days (bottom), including the gating strategy. Dotted histogram represents CD96 fluorescence minus one (FMO) control cells. (**B**) Expression of CD69 in viable CD4^+^ and CD8^+^ T cells activated for 3 days as assessed by flow cytometry. Data in (**A**) are representative of 2 independent experiments. Data in (**B**) are representative of T cells from at least 4 independent donors assessed in at least 3 independent experiments.

**Figure 2 cells-12-00309-f002:**
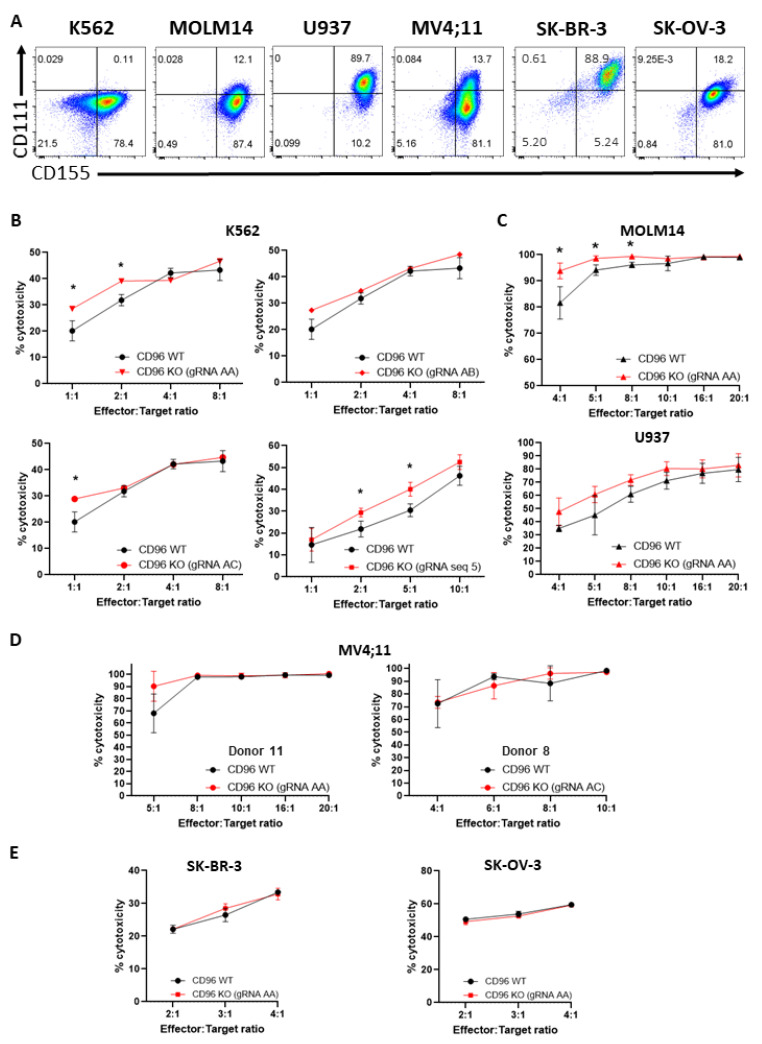
*CD96* deficiency in T cells enhances T cell cytotoxicity against K562 and MOLM14 cells. (**A**) Expression of CD96 ligands, CD155 and CD111, on K562, MOLM14, U937, MV4;11, SK-BR-3, and SK-OV-3 cells as assessed by flow cytometry. Quadrant gates were applied based on no staining for CD155 and CD111. (**B**) Percentage (%) cytotoxicity of T cells (calculated as described in Materials and Methods) against luciferase-expressing K562 cells 20 h following co-incubation of T and K562 cells at indicated E:T ratios. Each graph depicts T cells that were electroporated with different gRNAs. (**C**) % cytotoxicity of T cells against luciferase-expressing MOLM14 (top) and U937 (bottom) co-cultured with T cells as in (**B**). (**D**) % cytotoxicity of T cells, derived from 2 different human donors, against luciferase-expressing MV4;11 cells co-cultured with T cells as in (**B**). (**E**) % cytotoxicity of T cells against luciferase-expressing SK-BR-3 (left) and SK-OV-3 (right) co-cultured with T cells as in (**B**). Data in (**B**–**E**) are the mean ± SD of 3 technical replicates; multiple unpaired Student’s *t*-tests, *, *p* < 0.05. All data are representative of 2 independent experiments.

**Figure 3 cells-12-00309-f003:**
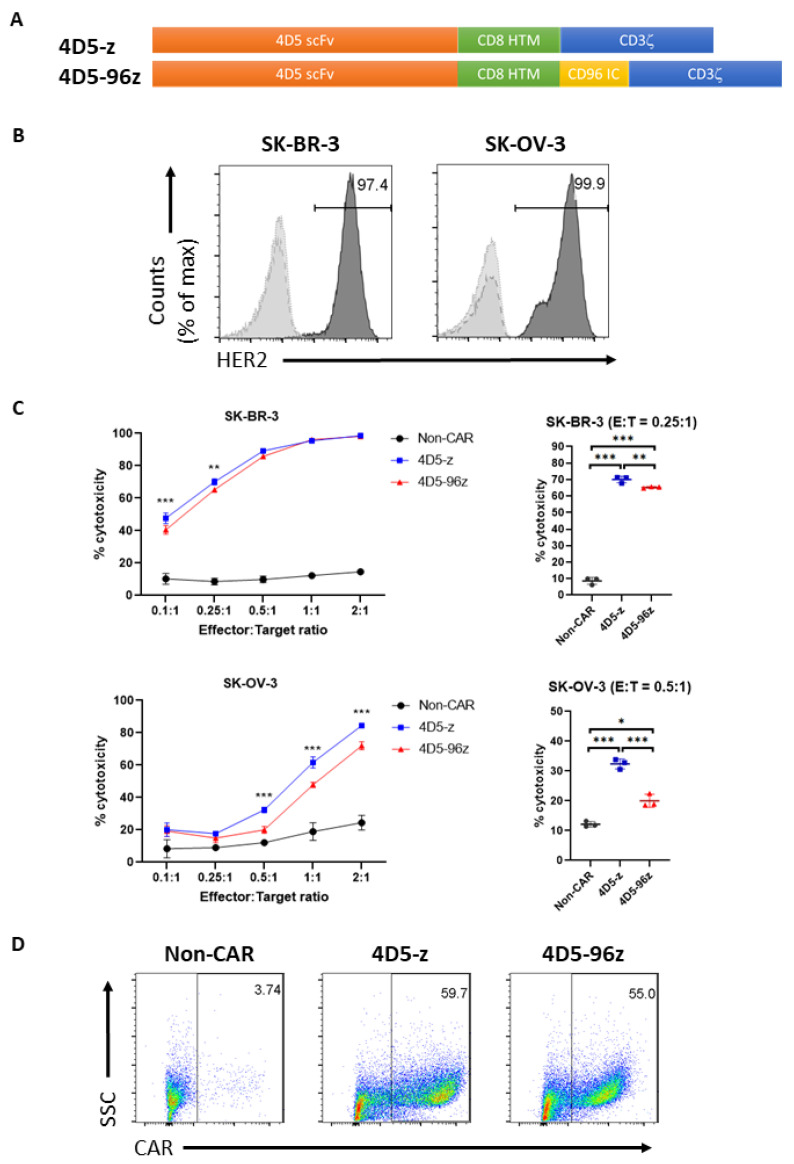
Inclusion of CD96 endodomain in 4D5-z CAR-T cells cripples their cytotoxicity against HER2^+^ tumor cells in vitro. (**A**) Schematic diagram of 4D5-z and 4D5-96z CAR constructs. scFv, single chain variable fragment; HTM, hinge and transmembrane domain; IC, intracellular signaling domain. (**B**) Expression of HER2 on SK-BR-3 and SK-OV-3 cells as assessed by flow cytometry. Dashed and dotted histograms represent unstained and isotype control cells, respectively. (**C**) Percentage (%) cytotoxicity of CAR-T cells against luciferase-expressing SK-BR-3 and SK-OV-3 cells 20 h after co-culture with T cells at indicated E:T ratios (left panels). For clarity, only statistical significance resulting from comparisons of % cytotoxicity of 4D5-z and 4D5-96z CAR-T cells are shown. Dot plots (right panels) show % cytotoxicity of non-CAR and CAR-T cells against SK-BR-3 and SK-OV-3 cells at E:T of 0.25:1 and 0.5:1, respectively. (**D**) Efficiency of CAR transduction as assessed by flow cytometry. Quadrant gates were applied based on CAR staining for non-CAR-T cells. Data in (C) are the mean ± SD of 3 technical replicates; two-way ANOVA with Tukey *post hoc* analysis, *, *p* < 0.05; **, *p* < 0.005; ***, *p* < 0.001. Data in (**C**,**D**) are representative of 2 independent experiments using 3 different donors.

**Figure 4 cells-12-00309-f004:**
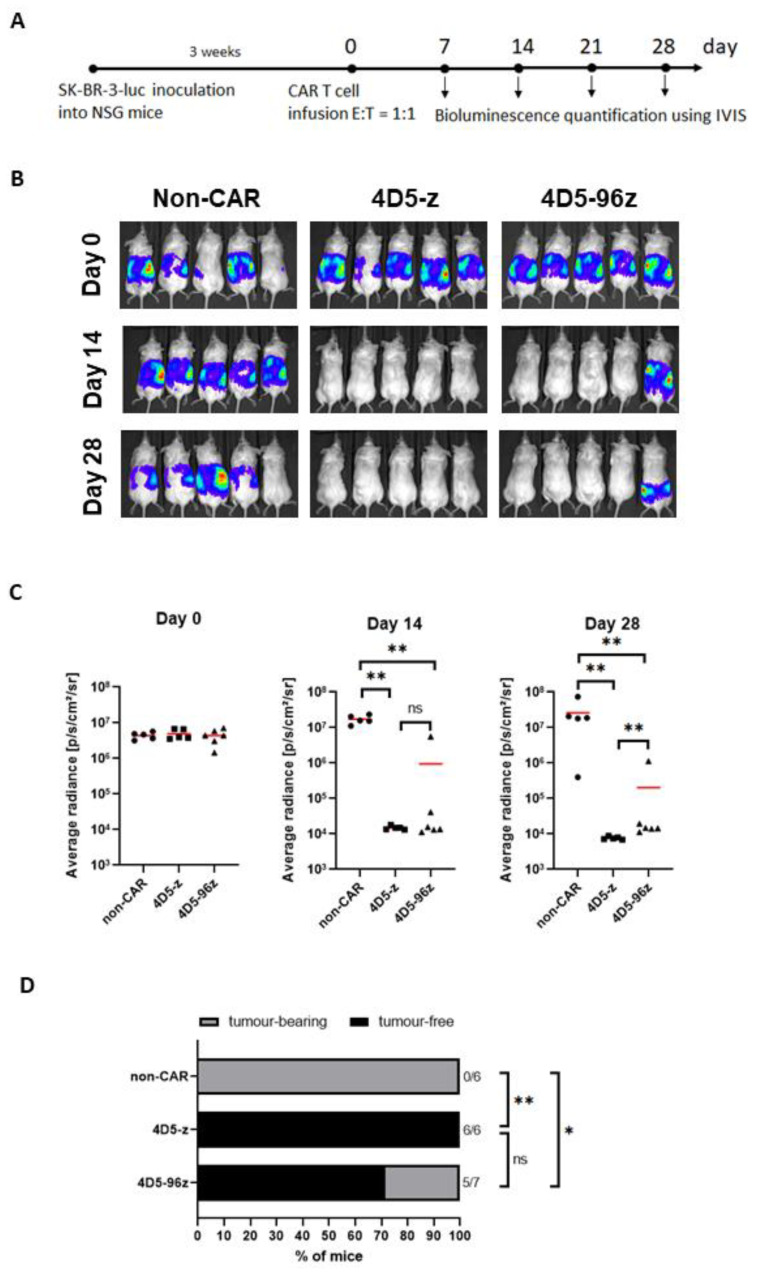
Incorporation of CD96 endodomain in 4D5-z CAR-T cells impairs their cytotoxicity against HER2^+^ tumor cells in vivo. (**A**) Schematic diagram of the tumor model using SK-BR-3 cells in NSG mice and treatment with non-CAR-T or CAR-T cells. (**B**) Selected images of tumor bioluminescence (BLI) at indicated time points (days) prior to and post T cell infusion. (**C**) Average radiance of tumor burden in mice at indicated time points post infusion of non-CAR-T, 4D5-z or 4D5-96z CAR-T cells. (**D**) Ratio of tumor-free and tumor-bearing mice of all mice examined at 14 days post T cell infusion. Data in (**B**,**C**) are representative of 2 independent experiments. Data in (**C**) are based on 5 or more mice analyzed with each symbol representing one mouse and red horizontal bars indicating the mean. Data in (**D**) are pooled from mice examined in 2 independent experiments; Kruskal-Wallis test (**C**) or Fisher exact test (**D**), *, *p* < 0.05; **, *p* < 0.005.

## Data Availability

The data presented in this study are available within the article and its Appendix A.

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
