# Peer review of "An Inhibitory Role for Human CD96 Endodomain in T Cell Anti-Tumor Responses"

_cells, 2023, doi:10.3390/cells12020309_

Round 1
Reviewer 1 Report
In this article, firstly the authors confirmed the role of CD96 in human T cells using CRISPR/Cas9 method. Moreover, the authors constructed T cells engineered with a chimeric antigen receptor (CAR) comprising human epidermal growth factor receptor 2 (EGFR2/HER2)-binding extracellular region and intracellular regions of CD96 and CD3z(4D5-96z CAR-T cells). Together, these findings implicate a role for CD96 endodomain in attenuating T cell cytotoxicity and support combination tumor immunotherapy targeting multiple rather than single immune checkpoints.
However, the whole experimental design is relatively simple. All the results are phenomena, and lack of essential mechanism research.
1. For the test of T cells cytotoxicity, except the results of % cytotoxicity of T cell mentioned in the article, the authors should test some cell cytokines.
2. It is well known that T cell activation is a continuous process with the different protein markers in the different activation states. In Figure S1, the authors just detected CD96 in the early T cell activation. The authors should increase CD96 in the whole process of T cell activation.
3. For the experiments in mice, the results only include tumor BLI. The authors should increase other data, such as tumor weight/size, body weight or the cytotoxicity of T cell from the tumor.
4. At first, the authors demonstrated that genetic ablation of CD96 in human T cells enabled these cells to kill chronic myeloid leukemia (CML) cells and a subset of acute myeloid leukemia (AML) cells more efficiently in vitro. However, the CAR-T (4D5-96z CAR-T cells) aimed to Her2-expression tumor. What is the relation between the two parts? The authors should talk about it in Discussion.
5. The authors found the inhibitory role for human CD96 endodomain in T cell tumor response. Then what is the specific mechanism?
Author Response
Dear Reviewer,
Thank you for reviewing our manuscript. Please find our response to your comments in the attached PDF file. Thank you.

Reviewer 2 Report
In the manuscript “An inhibitory role for human CD96 endodomain in T cell anti-tumor responses” the authors report their findings on the expression and function of CD96 in human T cells with respect to its putative function as an immune checkpoint molecule. The article is well-written, experimentally sound, conclusions are backed up by data and the topic is of general interest to a broad readership. However, there are a few issues that should be addressed to further improve the manuscript.
Data on the expression of CD96 in T cells is vital information and should not be provided as supporting information only
- Fig. 1A: Quadrant gates are set based on a no staining control and quadrants seem to artificially cut through populations thus hampering data interpretation. Moreover, did the authors apply an isotype control staining?
- The effects on killing do not seem to correlate with the efficiency of CD96 downregulation entirely. Did the authors try to block the interaction via antibodies? Especially with respect to therapeutic intervention this might be a more forward approach and might even lead to more consistent results.
- “In contrast, the loss of CD96 in T cells did not accentuate their killing of MV4;11, SK-BR-3 and SK-OV-3 cells regardless of E:T ratio and donor source of T cells (figures 1D and E), raising the possibility that ligands apart from CD155 and CD111 mediate the functional effects of CD96”. I do not completely understand this statement because the knock-down affects CD96 expression and the putative presence of additional ligating molecules should not have an influence when CD96 is absent.
Author Response

(The authors gave the same response as above.)

Round 2
Reviewer 1 Report
I have no more suggestions.
Author Response
Dear Reviewer,
We would like to thank you for taking time and effort to review our manuscript. We sincerely appreciate your valuable comments and suggestions, which helped us to improve the quality of our manuscript.
Thank you very much.